# National Scientific Funding for Interdisciplinary Research: A Comparison Study of Infectious Diseases in the US and EU

**Yoseob Heo [1,2]** , **Jongseok Kang [3] and Keunhwan Kim [2,*]**

[1] Department of Science and Technology Management and Policy, Korea University of Science and Technology (UST-Korea), 217, Gajeong-ro, Yuseong-gu, Daejeon 34113, Korea

[2] Korea Institute of Science and Technology Information (KISTI), Division of Data Analysis, 66, Hoegi-ro, Dongdaemun-gu, Seoul 02456, Korea

[3] Korea Institute of Science and Technology Information (KISTI), Busan Branch, CSP 4F, 79, Centum jungang-ro, Haeundae-gu, Busan 48058, Korea

* Correspondence: khkim75@kisti.re.kr; Tel.: +82-02-3299-6072

**Abstract:** Infectious diseases have been continuously and increasingly threatening human health and welfare due to a variety of factors such as globalisation, environmental, demographic changes, and emerging pathogens. In order to establish an interdisciplinary approach for coordinating R&D via funding, it is imperative to discover research trends in the field. In this paper, we apply machine learning methodologies and network analyses to understand how the European Union (EU) and the United States (US) have invested their funding in infectious diseases research utilising an interdisciplinary approach. The purpose of this paper is to use public R&D project data as data and to grasp the research trends of epidemic diseases in the US and EU through scientometric analysis.

**Keywords:** infectious diseases; United States; European Union; interdisciplinary; funding data

## 1. Introduction

Infectious diseases have been continuously and increasingly threatening human health and welfare due to a variety of factors such as globalisation, environmental, demographic changes, and emerging pathogens [1–3]. According to a study by the World Bank [4], the estimated economic losses from six major outbreaks, i.e., Nipah Virus (Malaysia), West Nile Fever (USA), SARS (Asia, Canada, other), HPAI (Asia, Europe), BSE (US, UK), and Rift Valley Fever (Tanzania, Kenya, Somalia) between 1997 and 2009, totalled $ 80 billion USD. If these outbreaks had been forestalled, an average of USD 6.7 billion per year of losses might have been avoidable, as indicated by the emergence and dissemination of the recent outbreak of the Ebola virus between 2013–2015, which resulted in around 28,600 suspected cases, 11,300 confirmed deaths and an estimated financial loss of USD 600 million worldwide [5]. Not only have infectious diseases directly and indirectly affected public health systems, but they have also influenced a variety of environmental (i.e., phenology) and economic (i.e., transportation industry) sectors [5,6]. Thus, a variety of researchers (i.e., molecular biologists, biochemists, epidemiologists) must collaborate closely to guarantee the transfer and application of scientific and technological outcomes to the field, thereby accomplishing long-term control of infectious diseases [1,7]. That is, the only solution would be internationally coordinated, interdisciplinary approach [8–12]. Such an argument can be identified in much medical research [13,14]. Also, diverse health-related databases have been used to gain opportunities for a better understanding of health care management across countries that differentiate from their health systems [15], or they may be used find a better direction for approaches to treatment [16,17].

In order to establish an interdisciplinary approach for coordinating R&D via funding, it is necessary to recognise the current research trends in the field [17,18]. Previous research on infectious diseases has focused on a scientific activity (i.e., publications) [1,19]. However, quantitative analyses based on publications or patents have an inherent limitation of retrospective characteristics [20]. Namely, it is inappropriate to use publications or patents to establish future-oriented strategies. As a consequence, prominent scholars have emphasised the utilisation of funding data as an alternative [20,21]. Although research based on funding was conducted, its focus was not the investigation of a disciplinary approach, but the distribution of funding for infectious diseases [22]. Thus, the first requirement was to research infectious diseases with both an interdisciplinary perspective and information from funding databases.

Moreover, 2015 was a year when the most research was conducted for these studies. Due to the Ebola breakout, The World Health Organization (WHO), a control centre of globally proactive and coordinated research and development (R&D) efforts, has pursued its core responsibility that averts and minimises the loss of life and economic resources stemming from an outbreak within its member states [9]. Thus, the second requirement for researchers and policymakers to understand is how we have dealt with the global challenge of infectious diseases since 2015.

To the best of our knowledge, no study has met both requirements. Therefore, in this study, we aimed to provide funding information on the interdisciplinary approach of infectious diseases since 2015. Many studies have indicated that the US and EU have critical roles in the scientific and technological advancements of infectious diseases [19,22]. In particular, as three leading national scientific funding organisations or programmes related to health domain, the National Institutes of Health (NIH) of the US and the framework programmes for research and innovation (i.e., Horizon 2020) funded by the EU are significant. The programmes of these organisations have emphasised fostering interdisciplinary studies among scientific disciplines such as life sciences, technical sciences and social sciences to keep their societies better because societal challenges such as health and wellbeing are relevant to multiple disciplines [21,23].

In this paper, we apply machine learning methodologies and network analyses to understand how the EU and US have invested their funding to address infectious diseases research using an interdisciplinary approach. Our research addresses the following questions:

1. What interdisciplinary fields of research on infectious diseases have the US and EU invested in since 2015?
2. What are the disciplinary ranges in infectious diseases-related research fields in the US and EU?
3. How does the US and the EU differ in their interdisciplinary research approaches to infectious diseases?

The remainder of this paper consists of four sections. Following this general introduction, the materials and methods section describes the framework and methodology. The results section presents the comparative results of the research profiling and machine learning analyses. The discussion and conclusion section review our research, identify research limitations and indicate promising research opportunities to pursue in the future.

## 2. Materials and Methods

### 2.1. Data Collection

The data used in this study are the global R&D project information, collected from the R&D database provided by the US and the EU. A total of 5934 R&D projects related to infectious diseases from 2014 to 2017 were collected from each database. The query set used to collect the data was the following: ((infectious OR contagious OR communicable) AND (disease *)). Data sources for each country and the number of data are shown in Figure 1.

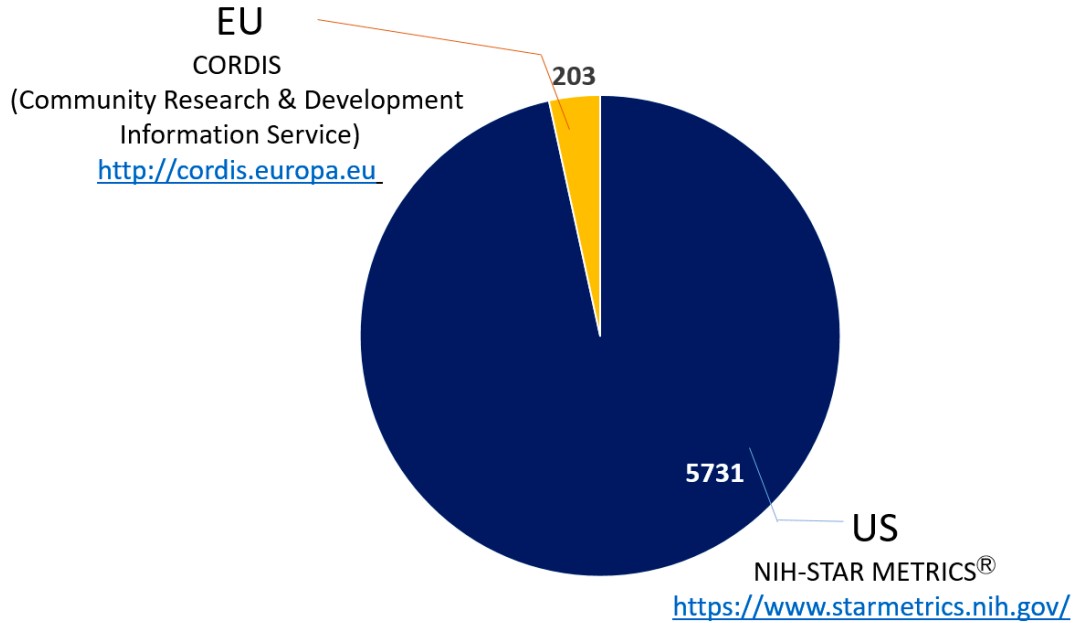

**Figure 1.** The source of global R&D project data.

STAR METRICS® is the U.S. federal government's database of R&D projects designed to create a data repository and tools needed to assess the impact of federal R&D investments. This database is funded by the NIH (National Institutes of Health), the NSF (National Science Foundation) and the U.S. Department of Agriculture and the Environment Department under the auspices of the Office of Science and Technology Policy (OSTP). Federal RePORTER is an initiative of STAR METRICS®, which creates a database of R&D projects from federal agencies and makes it available to the public. CORDIS is a major public database and portal site that provides the most extensive coverage of all major research projects, and the European Commission financially supports it. CORDIS's websites and repositories include all public information (project facts, publishable reports and outputs), communication and development assistance (news, events, success stories, magazines, etc.), open access publications and links to external sources held by the European Commission.

*2.2. Data Preprocessing*

Although the XML data provided by each country's R&D database had the same content, they could not be compared until there was a standardisation of the field names for each database. Accordingly, fields that contained the same contents but used different names for each database were unified into the same integrated field name by utilising the index of metadata provided by each database. Examples of uniformed field names are shown in Figure 2. The optimised database structure was completed by mapping the US and EU database fields into the unified field structure, and a new structured global R&D project database was completed when the actual field contents were parsed and filled in for the corresponding fields.

Also, in order to understand the convergence of the R&D area, a consistent classification system for each project was needed. As a result, 5 classification codes were assigned to each project by using the ASJC code (All Science Journal Classification Codes) of Scopus. The process of allocating ASJC codes to each project was done through the calculation of similarity through machine learning.

| Integrated field name | DB containing the field | Field contents | STAR METRICS® field | CORDIS field |
|---|---|---|---|---|
| PROJECT_SOURCE | ALL | Source of projects | 'FEDRIP' | 'CORDIS' |
| PROJECT_ID | ALL | Unique identifier number for each project | SM_APPLICATION_ID | RCN |
| PROJECT_DETAIL _SOURCE | ALL | Detailed source for each project | DEPARTMENT | FRAMEWORK _PROGRAMME |
| AGENCY | STAR METRICS® | Agency performing each project | AGENCY | - |
| IC_CENTER | STAR METRICS ® | IC center performing each project | IC_CENTER | - |
| GRANT_ID | ALL | Project identifier number managed by the project performing agency | PROJECT_NUMBER | REFERENCE |
| TITLE | ALL | Title of each project | PROJECT_TITLE | TITLE |
| ABSTRACT | ALL | Abstract of each project | ABSTRACT | OBJECTIVES |
| START_DATE | ALL | Project start date | PROJECT_START _DATE | START_DATE |

**Figure 2.** The examples of uniform field names.

The first step in the machine learning process is that the author keyword of approximately one million articles of Scopus and the ASJC codes assigned to each paper were set as the feature and label, respectively. After that, based on the similarity calculated according to the learned results, five ASJC codes that were most relevant to the title and abstract of the R&D project were given to each project. A conceptual diagram of this process is shown in Figure 3.

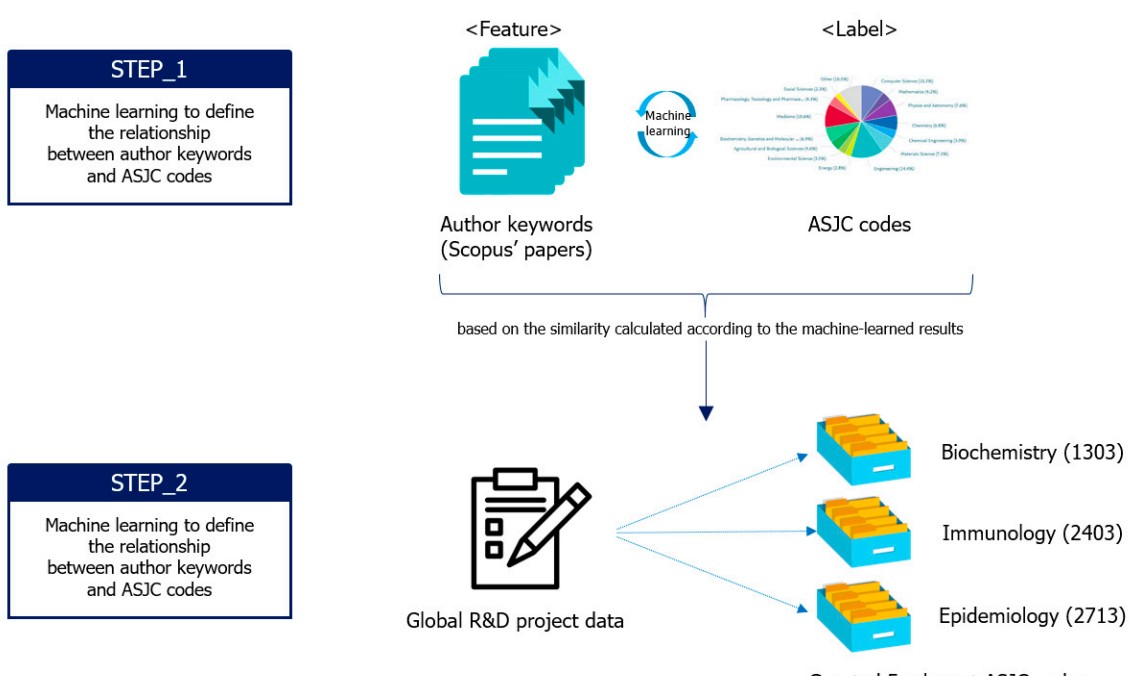

**Figure 3.** The process of assigning ASJC codes to global public R&D projects.

### 2.3. Cooccurrence Matrix

As a way to identify interdisciplinary R&D areas, we chose to validate project groups that were assigned to various areas, namely ASJC codes of various fields at the same time. As part of this approach, a cooccurrence matrix in text mining was used first. Cooccurrence in text mining refers to words appearing together in a sentence, paragraph or text. The ASJC codes that appear together in a particular project group will be relevant; the more often they appear, the greater their relevance. When multiple ASJC codes appear simultaneously in a project group, a network structure among ASJC codes is formed, and clusters composed of them are built, thereby enabling network analyses to be performed. Embodying and analysing the cooccurrence matrix was undertaken by using the Vantage Point® system (Search Tech, Inc., Atlanta, GA, USA, Version 7.1).

### 2.4. Clustering and Network Visualisation

As mentioned, when the association among ASJC codes is identified through the cooccurrence matrix, it has a network structure. By visualising this network structure, we can directly grasp the relationship between ASJC codes. The VOSViewer (Leiden University, Leiden, The Netherlands, Version 1.16.11) software was used as a network structure visualisation tool. The VOSViewer system calculates the similarity between each component and visualises the network structure in the form of a cluster map or a topographic map. A mathematical model and algorithm of the VOSViewer's clustering and mapping can be found in the research of Van Eck and Waltman [24].

Initially, the constructed clusters were able to divide into more detailed sub-clusters. Therefore, in order to derive more detailed, interdisciplinary R&D areas from larger clusters, the components of ASJC codes belonging to each cluster were extracted again, and each large cluster was divided into several sub-clusters through the two types of software, as mentioned earlier.

### 2.5. Defining an Interdisciplinary R&D Area

The definition of interdisciplinary R&D can only be seen by looking directly at the R&D projects that were comprised of actual clusters or sub-clusters. Therefore, we first ascertained the approximate R&D area by grasping the component ASJC codes constituting each sub-cluster. After that, the contents of the title and abstract of the project in the sub-cluster were checked, and the research fields of each sub-cluster were defined.

In addition, the budgets allocated to each sub-cluster and research institutes were identified. In order to compare countries, the US and EU projects were analysed individually by the same process, and through a comparison between the US and the EU, we sought to derive implications of the commonalities and differences between the US and the EU in interdisciplinary R&D areas related to infectious diseases.

## 3. Results

### 3.1. Interdisciplinary Research Areas on Infectious Diseases Funded by the US

As shown in Figure 4, the interdisciplinary research areas on infectious diseases that were funded by the US were divided into 5 clusters (categories); some of the clusters were then further categorised into 2~4 sub-clusters. After reviewing research descriptions of funded projects in each sub-cluster, we named each cluster to contain the comprehensive meaning of main research subjects as follows: The cluster of "Public health for HIV-vulnerable group/Respiratory health for children/HR for infectious diseases (Cluster 1)" is made of "HIV (sub-cluster 1-1)", "heath assessment of children (sub-cluster 1-2)", and "research scholars and educational programs (sub-cluster 1-3)", the cluster of "Diagnosis and treatment of infectious diseases by using advanced technology (Cluster 2)" is composed of "information technology based infectious disease diagnosis (sub-cluster 2-1)", "molecular biology based diagnosis and treatment (sub-cluster 2-2)", "wastewater treatment for preventing infectious diseases (sub-cluster 2-3)", and "eye disease stemming from infectious diseases (sub-cluster 2-4)", the cluster of "Biological

studies on the mechanism of inflammatory diseases caused by infectious diseases and development of therapies for them (Cluster 3)" is created by "inflammation treatments caused by infectious diseases (sub-cluster 3-1)", "sleeping sickness (sub-cluster 3-2)", and "viral diseases in human and animals (sub-cluster 3-3)", the cluster of "Strengthening research capacity for epidemiology (Cluster 4)" is formed by "small animal models for infectious diseases (sub-cluster 4-1)" and "epidemiology and health system (sub-cluster 4-2)", and the cluster of "Clinical trials on vaccines and products to help treat and prevent infectious diseases (Cluster 5)". In the next subsection, detailed investigations for each cluster will be described. Comprehensive information for each cluster of the US is listed in Table A1 of Appendix A.

### 3.1.1. Public Health for HIV-Vulnerable Group/Respiratory Health for Children/HR for Infectious Diseases (Cluster 1)

Public health for HIV-vulnerable group/Respiratory health for children/HR for infectious diseases (Cluster 1) contained 46 projects costing USD 9,270,970 worth of and 21 multiple disciplines, respectively (see Figure 5 and Table 1).

First, HIV (Sub-cluster 1-1) consisted of 34 projects totalling USD 4,351,252; they have ASJC codes such as Health (social science) (3306), Health Policy (2719), Nursing (miscellaneous) (2901) with different disciplines such as Oncology (2917), Psychiatric Mental Health (2921), Social Psychology (3207) and Communication (3315). For example, the Infectious Diseases Institute conducted the project, "HIV Self Testing to Empower Prevention Choices in Sex Workers", which is budgeted to spend USD 108,863 during the time period of 2017–2022. To accomplish the project, researchers came from multiple disciplines such as Social Psychology (3207), Health (social science) (3306), Health Policy (2719), Oncology(nursing) (2917), and Nursing (miscellaneous) (2901) have participated. For improving international health, Duke University has studied "Tanzania under the project title, Integrating Mental Health into an HIV Clinic to Improve Outcomes in Tanzanian Youth", with a budget of USD 135,050 during the time period of 2015–2020. Meanwhile, for the 2015–2019 time period, University of North Carolina-Chapel Hill has concentrated on prisoners who have been exposed to HIV in Zambia in their research of "Understanding Longitudinal Clinical Outcomes and Post-release Retention in Care among HIV-infected Prisoners in Lusaka Zambia", from the viewpoint of multiple disciplines such as Psychiatric Mental Health (2921), Social Psychology (3207), Health (social science) (3306), Health Policy (2719), and Nursing (miscellaneous) (2901).

Second, health assessments of children who were exposed to infectious disease at an early age, environmental pollutants, chemical exposure, and endocrine disruptors (Sub-cluster 1-2) is formed with USD 3,530,344 worth of 2 projects. Health (2739) played a fundamental role and adopted four heterogeneous research areas such as Epidemiology (2713), Obstetrics and Gynaecology (2729). The representative project, "Early Life Exposures and Child Trajectories Growth and Respiratory Health", which was supervised by the University of Utah, was accomplished through interdisciplinary approaches including Medicine (miscellaneous) (2701), Public Health, Environmental and Occupational Health (2739), Obstetrics and Gynaecology (2729), Epidemiology (2713), and Radiation (3108) with USD 2,112,977 from 2016–2018.

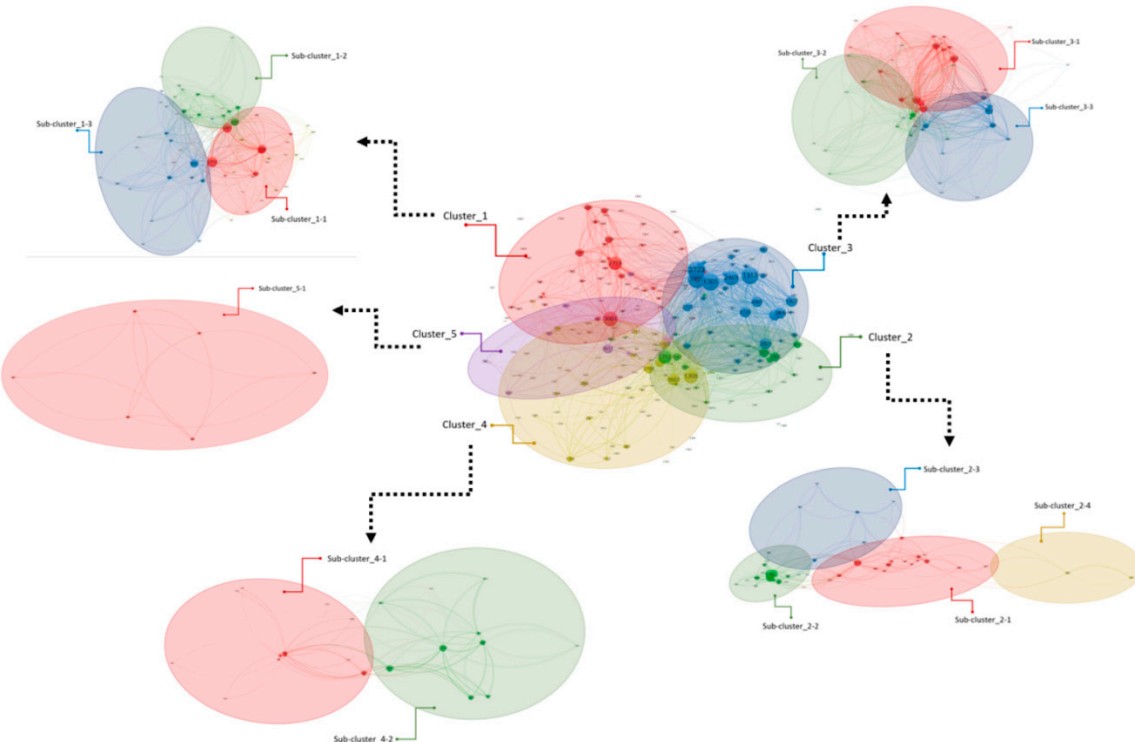

**Figure 4.** Clusters of interdisciplinary research on infectious diseases funded by the US.

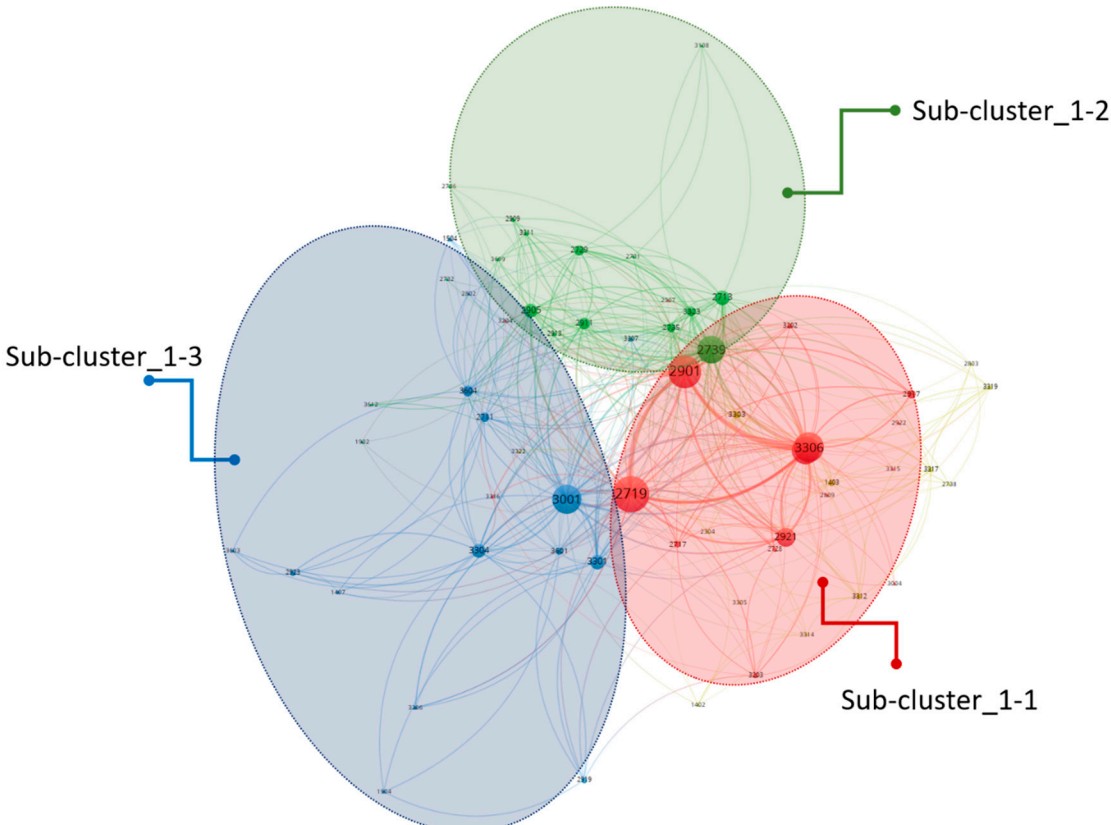

**Figure 5.** Public health for HIV-vulnerable group/ Respiratory health for children/HR for infectious diseases (Cluster 1).

**Table 1.** Public health for HIV-vulnerable group/ Respiratory health for children/HR for infectious diseases (Cluster 1).

| No | COST (USD) | ORG_NAME | ASJC_CODE_5 | START_DATE | END_DATE | TITLE |
|---|---|---|---|---|---|---|
| 1-1 | 136,046 | University of North Carolina Chapel Hill | 2921;2719;3306;3207;2901 | 2015-09-23 | 2019-08-31 | Understanding Longitudinal Clinical Outcomes and Post-release Retention in Care among HIV-infected Prisoners in Lusaka Zambia |
| 1-1 | 727,222 | Yale University | 2719;3306;2921;3207;2901 | 2016-08-15 | 2021-04-30 | Addiction HIV and Tuberculosis in Malaysian Criminal Justice Settings |
| 1-1 | 181,759 | University of Chicago | 3207;3306;2719;2917;2901 | 2015-04-01 | 2018-03-31 | Public Health Targeting of Prep at HIV Positives Bridging Networks |
| 1-1 | 108,863 | Infectious Diseases Institute | 3207;3306;2719;2917;2901 | 2017-08-22 | 2022-05-31 | HIV Self Testing to Empower Prevention Choices in Sex Workers |
| 1-2 | 2,112,977 | University of Utah | 2701;2739;2729;2713;3108 | 2016-09-21 | 2018-08-31 | Early Life Exposures and Child Trajectories Growth and Respiratory Health |
| 1-3 | 185,118 | Tufts University Boston | 3601;1904;2919;3301;3304 | 2013-08-15 | 2016-07-31 | Modelling for Fidelity Mentored Dissemination of a Novel Curriculum about Infection |
| 1-3 | 109,601 | Colorado State University-Fort Collins | 3001;2923;3304;3604;3603 | 2013-07-15 | 2018-06-30 | Veterinary Pre-doctoral Research Scholars Program |

Third, developing research and educational programs for fostering epidemic researchers (Sub-cluster 1-3) comprised 10 projects worth a total of USD 1,389,374, and coordinated nine different disciplines. The research areas of Pharmacology, Toxicology and Pharmaceutics (miscellaneous) (3001) played the main roles, while another eight disciplines including Social Sciences (miscellaneous) (3301) and Education (3304) also participated in completing the goals of this research subject. Representatively, the project "Veterinary Pre-doctoral Research Scholars Program", conducted by Colorado State University-Fort Collins with funding of USD 109,601 from 2013–2018, incorporated a variety of disciplines including Pharmacology, Toxicology and Pharmaceutics (miscellaneous) (3001), Review and Exam Preparation (2923), Education (3304), Emergency Medical Services (3604), Complementary and Manual Therapy (3603).

3.1.2. Diagnosis and Treatment of Infectious Diseases by Using Advanced Technology (Cluster 2)

Overall, the cluster of the diagnosis and treatment of infectious diseases by using advanced technology (Cluster 2) was comprised of 31 projects totalling USD 18,305,522 and incorporating 38 multiple disciplines, respectively (see Figure 6 and Table 2).

First, based on Health Informatics (2718), information technology-based infectious disease diagnosis (sub-cluster 2-1) collaborated with 11 heterogeneous disciplines such as Health Information Management (3605), Computer Networks and Communications (1705), Hardware and Architecture (1708) and Biomedical Engineering (2204). Nineteen projects spent USD 5,532,582. In particular, the University of Pittsburgh spent USD 2,330,227 from 2014~2019 for the project of "MIDAS (Modelling of Infectious Disease Agent Study) Informatics Services Group ISG" and the University of Utah is estimated to spend USD 747,704 from 2017~2021 in the project of "IOBIO Web-based Interactive Tools for Real-time Analysis in Genomic Big Data".

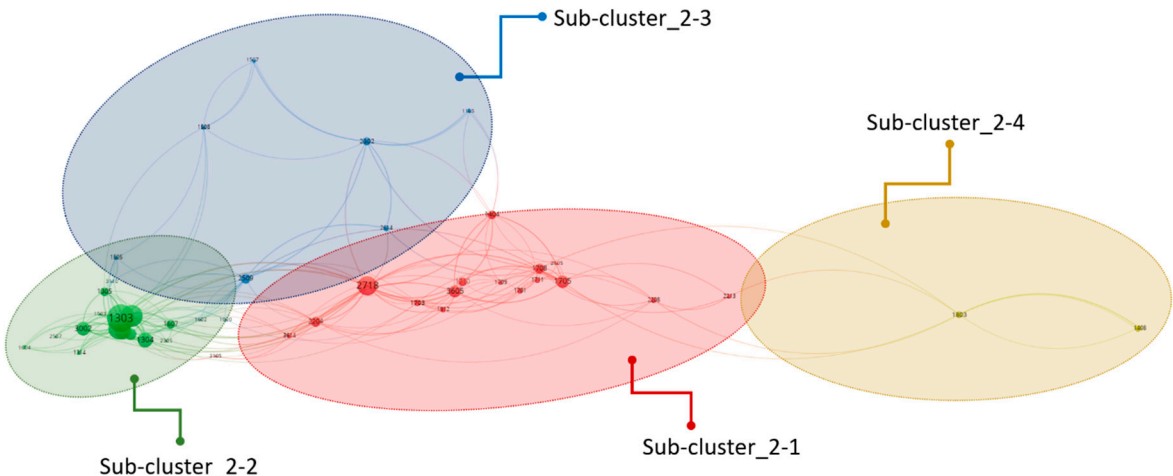

**Figure 6.** Diagnosis and treatment of infectious diseases by using advanced technology (Cluster 2).

Second, there were 19 projects in the sub-cluster 2-2 of the molecular biology-based diagnosis and treatment of infectious diseases though, spending USD 5,532,582, based on the discipline of Biochemistry (1303), and collaborating with 14 different disciplines, such as Biophysics (1304), Biotechnology (1305), Spectroscopy (1607) and Drug Discovery (3002). The project, "Combinatorial Biosynthesis of Fungal Benzene diol Lactone Polyketides", conducted by the University of Arizona with USD 289,273 during the time period of 2016–2020, is a typical example. It covers Biochemistry (1303), Drug Discovery (3002), Structural Biology (1315), Pharmaceutical Science (3003) and Biophysics (1304). Another example is the project, "Structural Transitions in Proteins and Protein Assemblies", studied by the University of Oklahoma with a budget of USD 296,900 from 2017–2021 consisting of multiple approaches such as Biochemistry (1303), Pharmaceutical Science (3003), Biophysics (1304), Spectroscopy (1607) and Structural Biology (1315).

**Table 2.** Diagnosis and treatment of infectious diseases by using advanced technology (Cluster 2).

| No. | COST (USD) | ORG_NAME | ASJC_CODE_5 | START_DATE | END_DATE | TITLE |
|---|---|---|---|---|---|---|
| 2-1 | 2,330,227 | The University of Pittsburgh | 3605;2718;1710;1705;1404 | 2014-08-05 | 2019-04-30 | Modelling of Infectious Disease Agent Study(MIDAS) Informatics Services Group ISG |
| 2-1 | 747,704 | University of Utah | 3605;2718;2204;1710;3614 | 2017-08-01 | 2021-05-31 | IOBIO Web-based Interactive Tools for Real-time Analysis in Genomic Big Data |
| 2-1 | 99,552 | Research Triangle Institute | 2718;3605;1705;1701;1710 | 2014-06-01 | 2014-09-30 | MIDAS ITR Extension |
| 2-2 | 289,273 | University of Arizona | 3002;1315;1303;3003;1304 | 2016-04-01 | 2020-03-31 | Combinatorial Biosynthesis of Fungal Benzene diol Lactone Polyketides |
| 2-2 | 296,900 | University of Oklahoma | 3003;1303;1304;1607;1315 | 2017-09-15 | 2021-08-31 | Structural Transitions in Proteins and Protein Assemblies |
| 2-3 | 332,441 | Rutgers State University of New Jersey—New Brunswick | 2302;1507;2308;1508;1909 | 2016-03-15 | 2019-02-28 | UNS Dynamics of Microbial Agents in Sewer Systems and Wet Weather Flow |
| 2-4 | 399,420 | University of California San Francisco | 2705;3504;2201;1803;1408 | 2016-05-01 | 2020-04-30 | Forecasting Trachoma Control |

Third, wastewater treatment for preventing infectious diseases is rooted in a variety of knowledge stemming from Strategy and Ecological Modelling (2302), Fluid Flow and Transfer Processes (1507),

Process Chemistry and Technology (1508), Geotechnical Engineering and Engineering Geology (1909), and Ecological Modelling (2302) Management, Monitoring, Policy and Law (2308). Its funding totalled USD 332,441. The "UNS Dynamics of Microbial Agents in Sewer Systems and Wet Weather Flow" was scrutinised by Rutgers, State University of New Jersey—New Brunswick from 2016–2019.

Fourth, eye diseases stemming from infectious diseases (i.e., Trachoma) comprises Management (1408), Management Science and Operations Research (1408), Engineering (miscellaneous) (2201), Cardiology and Cardiovascular Medicine (2705) and Oral Surgery (3504). The project, "Forecasting Trachoma Control", studied by University of California San Francisco with a budget of USD 399,420 from 2016–2019, can be typified.

3.1.3. Biological Studies on the Mechanism of Inflammatory Diseases Caused by Infectious Diseases and the Development of Therapies for Them (Cluster 3)

Biological studies on the mechanism of inflammatory diseases caused by infectious diseases and the development of therapies for them (Cluster 3) contained 131 projects worth USD 37,588,973 and spanned 35 multiple disciplines, respectively (see Figure 7 and Table 3).

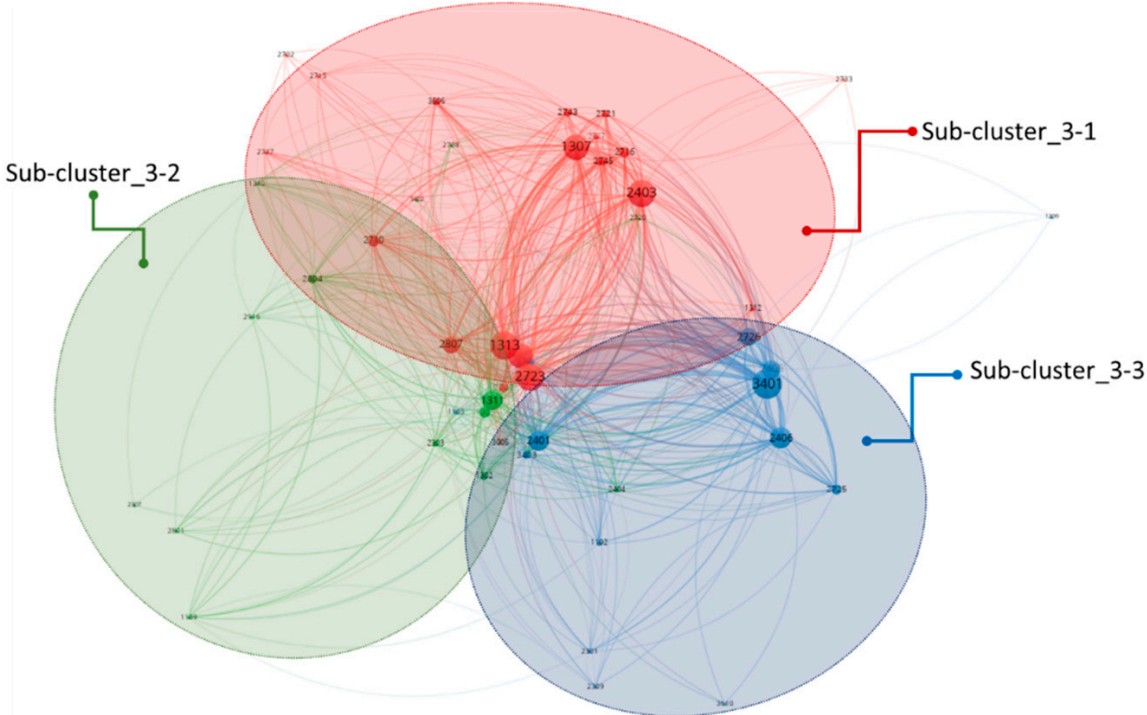

**Figure 7.** Biological studies on the mechanism of inflammatory diseases caused by infectious diseases and development of therapies for them (Cluster 3).

First, inflammation treatments caused by infectious diseases accounted for 103 projects receiving a total of USD 31,226,545 and comprising 21 heterogeneous disciplines. The principal three and collaborative 18 research areas were Cell Biology (1307), Immunology (2403), Immunology and Allergy (2723) and Embryology (2710), Rheumatology (2745), Endocrine and Autonomic System (2807), Genetics (clinical) (2716), etc. Regarding inflammation, The Harvard School of Public Health completed the project, "Elucidating a Role for Calcium Signalling in Activation of the Nlrp3 Inflammasome", with USD 379,525 from 2013–2018 in multiple disciplines such as Rheumatology (2745), Cell Biology (1307), Molecular Medicine (1313), Nephrology (2727) and Genetics (clinical) (2716). Moreover, the Children's Hospital Medical Center at Cincinnati and the University of Arizona carried out the project, "Prenatal Inflammatory Exposures and Neonatal Immune Development" with expected expenditure of USD 564,999, between 2017–2022, using interdisciplinary approaches including the disciplines of Cell Biology

(1307), Reproductive Medicine (2743), Endocrine and Autonomic Systems (2807), Embryology (2710), Immunology and Allergy (2723), and "Modulation of Dendritic Cell Function in the Pathogenesis of Inflammatory Bowel Diseases" (USD 345,375, during 2016–2021), interdisciplinary approaches utilised research from Cell Biology (1307), Endocrine and Autonomic Systems (2807), Immunology (2403), Molecular Medicine (1313) and Embryology (2710). Meanwhile, for mechanism, the Mayo Clinic has concentrated on the project, "Epigenetic Mechanisms Underlying Hepatitis C-induced Hepatocarcinogenesis", which has been allocated USD 620,487 between 2017 and 2021, combining the disciplines of Hepatology (2721), Cell Biology (1307), Genetics (clinical) (2716), Molecular Medicine (1313) and Gastroenterology (2715). The Columbia University of Health Sciences' project, "Mechanisms Mediating Tissue Repair by Leukocytes during Influenza Virus Infection", had a budget of USD 161,400 during 2017–2019 on the basis of multiple disciplines such as Cell Biology (1307), Molecular Medicine (1313), Nephrology (2727), Reproductive Medicine (2743) and Immunology (2403).

**Table 3.** Biological studies on the mechanism of inflammatory diseases caused by infectious diseases and development of therapies for them (Cluster 3).

| No | COST (USD) | ORG_NAME | ASJC_CODE_5 | START_DATE | END_DATE | TITLE |
|---|---|---|---|---|---|---|
| 3-1 | 545,205 | H Lee Moffitt Cancer Center & Research Institute | 2747;1307;1313;2710;2723 | 2013-08-01 | 2018-04-30 | Adoptive Transfer of Donor Tregs Specific against Host Alloantigens for Presention |
| 3-1 | 345,375 | University of Arizona | 1307;2807;2403;1313;2710 | 2016-09-15 | 2021-08-31 | Modulation of Dendritic Cell Function in the Pathogenesis of Inflammatory Bowel Diseases |
| 3-1 | 620,487 | Mayo Clinic | 2721;1307;2716;1313;2715 | 2017-09-30 | 2021-06-30 | Epigenetic Mechanisms Underlying Hepatitis C-induced Hepatocarcinogenesis |
| 3-1 | 564,999 | Children's Hospital Medical Center at Cincinnati | 1307;2743;2807;2710;2723 | 2017-09-01 | 2022-08-31 | Prenatal Inflammatory Exposures and Neonatal Immune Development |
| 3-1 | 303,140 | University of Utah | 1307;1313;2737;1306;2723 | 2017-05-15 | 2018-11-14 | Respiratory Immune Dysregulation Following Intestinal Infection |
| 3-1 | 455,611 | University of Alabama at Birmingham | 1307;2716;2723;2807;2747 | 2017-03-01 | 2022-02-28 | Specialization of Innate and Adaptive Immune Cells in Intestinal Barrier Function |
| 3-2 | 416,250 | Yale University | 2804;1309;1109;1311;2303 | 2014-03-15 | 2019-02-28 | Mechanism of Infectivity Acquisition in African Trypanosomes |
| 3-3 | 227,450 | University of California Davis | 3401;2405;2406;2725;2726 | 2016-08-16 | 2018-07-31 | Mechanisms of Increased Susceptibility to Salmonella Colonization during Malaria |
| 3-3 | 349,968 | University of Washington | 3401;2301;2405;2726;2725 | 2017-08-16 | 2020-05-31 | Impact of Malaria Coinfection on HIV Vaccination |
| 3-3 | 895,500 | Agricultural Research Service | 2401;2406;3403;2301;2726 | 2014-04-17 | 2017-04-30 | Predictive Biology of Emerging Vector-borne Viral Diseases |

Second, with regard to sleeping sickness (i.e., Trypanosoma brucei), a total of four projects were selected, and USD 416,250 worth of funding was allocated. At least four heterogeneous fields, i.e., Insect Science (1109), Ecology (2303), Cellular, Molecular Neuroscience (2804), etc. have collaborated with Genetic (1311) as a fundamental discipline. With expenditure of USD 416,250 during the 2014–2019 time period, Yale University's project, "Mechanism of Infectivity Acquisition in African Trypanosomes" was exemplified as an interdisciplinary approache which included the disciplines of Insect Science

(1109), Developmental Biology (1309), Genetics (1311), Ecology (2303) and Cellular and Molecular Neuroscience (2804).

Third, USD 5,946,178 was spent on 24 projects studying viral diseases in human and animals, which consisted of various disciplines such as Virology (2406), Veterinary (miscellaneous) (3401), Immunology and Microbiology (miscellaneous) (2401), along with seven heterogeneous research fields including Infectious diseases (2725), Microbiology (medical) (2726) and Parasitology (2405). For instance, the University of California Davis's research project, "Mechanisms of Increased Susceptibility to Salmonella Colonization during Malaria", with interdisciplinary approaches such as Veterinary (miscellaneous) (3401), Parasitology (2405), Virology (2406), Infectious Diseases (2725) and Microbiology (medical) (2726), spent USD 227,450 during the time period of 2016–2018. Furthermore, the Agricultural Research Service's project, "Predictive Biology of Emerging Vector-borne Viral Diseases", collaborated among multiple disciplines including Immunology and Microbiology (miscellaneous) (2401), Virology (2406), Food Animals (3403), Environmental Science (miscellaneous) (2301) and Microbiology (medical) (2726) between 2014–2017 and spent USD 895,500.

3.1.4. Strengthening Research Capacity for Epidemiology (Cluster 4)

Generally strengthening research capacity for epidemiology (Cluster 4) comprised USD 113,924,725 worth of 121 projects and 19 multiple disciplines, respectively (see Figure 8 and Table 4).

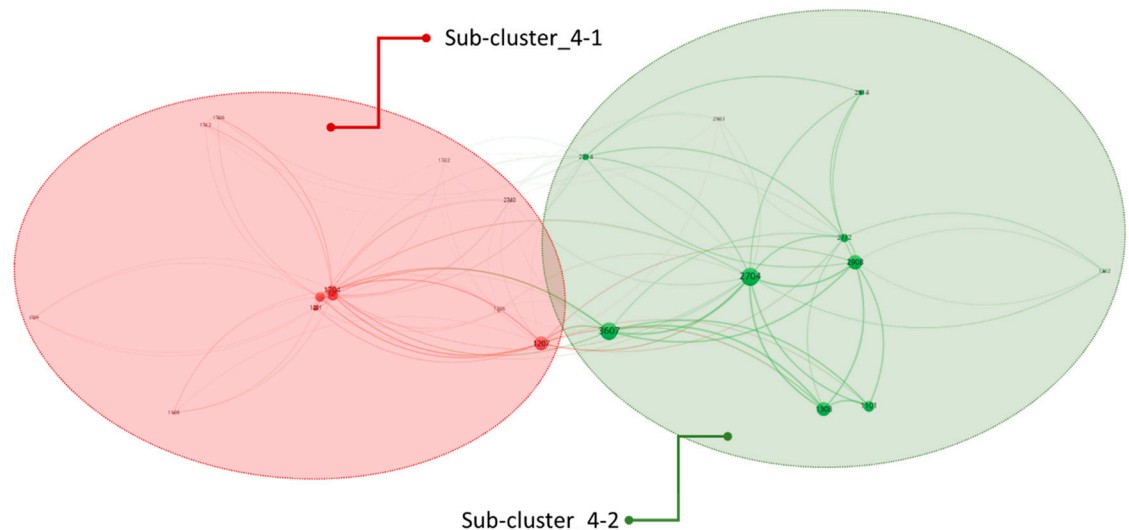

**Figure 8.** Strengthening research capacity for epidemiology (Cluster 4).

First, the small animal models for infectious diseases consisted of 19 projects with spending of USD 8,688,987, which principally studied Computer Graphics and Computer-Aided Design (1704). Art and Humanities (miscellaneous) (1201), History and Philosophy of Science (1207), Software (1712), etc. were coordinated. The project, "Woodchuck Animal Model for Testing Hepatitis B Virus Therapeutics", was researched by the Georgetown University with a budget of USD 1,080,000 between 2016–2017 collaborated with the disciplines of Computer Graphics and Computer-Aided Design (1704), Arts and Humanities (miscellaneous) (1201), Mathematics (miscellaneous) (2601), Horticulture (1108) and Complementary and alternative medicine (2707). "Small Animal Models for Biodefense Viruses" published by the Utah State University with a budget of USD 813,138 between 2017–2018 utilised the disciplines of Computer Graphics and Computer-Aided Design (1704), Mathematics (miscellaneous) (2601), Horticulture (1108), History and Philosophy of Science (1207) and Arts and Humanities (miscellaneous) (1201).

**Table 4.** Strengthening research capacity for epidemiology (Cluster 4).

| No. | COST (USD) | ORG_NAME | ASJC_CODE_5 | START_DATE | END_DATE | TITLE |
|---|---|---|---|---|---|---|
| 4-1 | 1,080,000 | Georgetown University | 1704;1201;2601;1108;2707 | 2016-04-12 | 2017-03-14 | Task D13 Woodchuck Animal Model for Testing Hepatitis B Virus Therapeutics |
| 4-1 | 813,138 | Utah State University | 1704;2601;1108;1207;1201 | 2017-01-11 | 2018-01-10 | Task A105 Small Animal Models for Biodefense Viruses |
| 4-2 | 656,961 | Louisiana State Office of Public Health | 1308;2704;3607;2908;1101 | 2014-08-01 | 2019-07-31 | Epidemiology and Laboratory Capacity for Infectious Diseases (ELC)—Building and Strengthening Epidemiology Laboratory and Health Information Systems Capacity in State and Local Health Departments |
| 4-2 | 2,242,886 | New York City Health/Mental Hygiene | 2704;2908;2712;2714;2914 | 2014-08-01 | 2019-07-31 | Nycdohmh Pphf 2014 Epidemiology and Laboratory Capacity for Infectious Diseases (ELC) |
| 4-2 | 487,936 | Virgin Islands Department of Health | 2704;2908;2712;2714;2914 | 2014-08-01 | 2019-07-31 | Epidemiology and Laboratory Capacity for Infectious Diseases |

Second, 102 projects in the field of epidemiology and health systems spent a total of USD 105,235,738, principally based on disciplines such as Biochemistry, medical (2704), Fundamentals and skills (2908) and Medical Laboratory Technology (3607) while collaborating with Agricultural and Biological Sciences (miscellaneous) (1011), Clinical Biochemistry (1308), Endocrinology, Diabetes, and Metabolism (2712), Family Practice (2714), Medical-Surgical (2914) and so on. Examples of these projects include "Epidemiology and Laboratory Capacity for Infectious Diseases (ELC)—Building and Strengthening Epidemiology Laboratory and Health Information Systems Capacity in State and Local Health Departments" studied by the Louisiana State Office of Public Health with a budget of USD 656,961 between 2014–2019 and incorporating multiple disciplines including Clinical Biochemistry (1308), Biochemistry, medical (2704), Medical Laboratory Technology (3607), Fundamentals and skills (2908) and Agricultural and Biological Sciences (miscellaneous) (1011). Nycdohmh Pphf 2014 Epidemiology and Laboratory Capacity for Infectious Diseases (ELC) was conducted by the New York City Health/Mental Hygiene with a budget of USD 2,242,886 between 2014–2019, and utilised the research of multiple disciplines including Biochemistry, medical (2704), Fundamentals and skills (2908), Endocrinology, Diabetes and Metabolism (2712), Family Practice (2714) and Medical-Surgical (2914). "Epidemiology and Laboratory Capacity for Infectious Diseases" investigated by the Virgin Islands Department of Health, spending USD 487936 during the time period of 2014–2019 with various disciplines including Biochemistry, medical (2704), Fundamentals and skills (2908), Endocrinology, Diabetes and Metabolism (2712), Family Practice (2714) and Medical-Surgical (2914).

3.1.5. Clinical Trials on Vaccines and Products to Help Treat and Prevent Infectious Diseases (Cluster 5)

Regarding clinical trials of vaccines and products to help treat and prevent infectious diseases, two projects totalled USD 80,156 which were conducted through the following disciplines: Critical Care and Intensive Care Medicine (2706), Industrial relations (1410) and Applied Mathematics (2604) (see Figure 9 and Table 5).

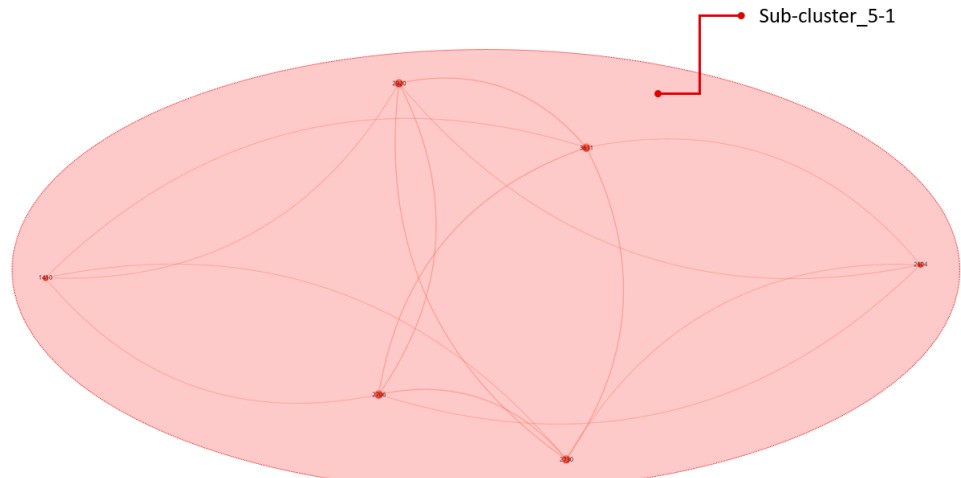

**Figure 9.** Clinical trials on vaccines and products to help treat and prevent infectious diseases (Cluster 5).

**Table 5.** Clinical trials on vaccines and products to help treat and prevent infectious diseases (Cluster 5).

| No. | COST (USD) | ORG_NAME | ASJC_CODE_5 | START_DATE | END_DATE | TITLE |
|---|---|---|---|---|---|---|
| 5-1 | 26,886 | Saint Louis University | 3611;2920;2730;1410;2706 | 2016-07-12 | 2017-07-11 | Vaccine and Treatment Evaluation Unit Clinical Operations Support for Influenza Vaccine Studies |
| 5-1 | 53,270 | University of Iowa | 3611;2730;2920;2706;2604 | 2015-07-13 | 2016-10-01 | VTEU Clinical Trial Operations Support |

The Saint Louis University's project, "Vaccine and Treatment Evaluation Unit Clinical Operations Support for Influenza Vaccine Studies", included the interdisciplinary viewpoints of Pharmacy (3611), Oncology (2730), Pharmacology (nursing) (2920), Critical Care and Intensive Care Medicine (2706), and Industrial relations (1410) during the time period of 2016–2017 and spent USD 26,886. In addition, the University of Iowa's project, "VTEU Clinical Trial Operations Support", utilised multiple perspectives such as Pharmacy (3611), Oncology (2730), Pharmacology (nursing) (2920), Critical Care and Intensive Care Medicine (2706), Applied Mathematics (2604) during the 2015–2016 time period, spending USD 53,270.

## 3.2. Fields of Interdisciplinary Research on Infectious Diseases Funded by the EU

As shown in Figure 10, the interdisciplinary research areas on infectious diseases that were funded by the EU were divided into 4 clusters (categories). Comprehensive information for each cluster of the EU is listed in Table A2 of Appendix A.

### 3.2.1. Detection and Profiling for Pathogens (Cluster 1)

Five projects on the detection and profiling for pathogens were allocated USD 9,244,467, which principally studied Medical Laboratory Technology (3607) along with 15 different disciplines, including specialities such as Virology (2406), Biochemistry, medical (2704), Clinical Biochemistry (1308), Immunology and Microbiology (miscellaneous) (2401) (see Table 6).

**Table 6.** Detection and profiling for pathogens (Cluster 1).

| No. | COST (USD) | ORG_NAME | ASJC_CODE_5 | START_DATE | END_DATE | TITLE |
|---|---|---|---|---|---|---|
| 1 | 205,891 | Erasmus Universitair Medisch Centrum Rotterdam | 3403;2704;2401;3607;2406 | 2014-07-01 | 2016-06-30 | Early Detection of Emerging Viruses by Next Generation in Situ Hybridization |
| 1 | 83,572 | Millidrop Instruments SAS | 2726;2402;1308;1502;3607 | 2017-08-01 | 2018-01-31 | New Millidrop Analyzer Miniaturized and Faster Clinical Microbiology Testing in Only One Drop |
| 1 | 2,924,954 | University of Oslo | 2401;2718;1404;1207;2208 | 2017-08-01 | 2022-07-31 | Scalable Inference Algorithms for Bayesian Evolutionary Epidemiology |
| 1 | 4,042,309 | Cy.R.I.C Cyprus Research and Innovation Center Ltd. | 1308;2704;2726;3403;3607 | 2017-11-01 | 2021-04-30 | Swine Diseases Field Diagnostics Toolbox |
| 1 | 1,987,741 | Ecole Polytechnique Federale de Lausanne | 2401;1314;2406;1105;2721 | 2013-02-01 | 2018-01-31 | Virulence Factors of Facultative Pathogens and their Role outside the Human Host |

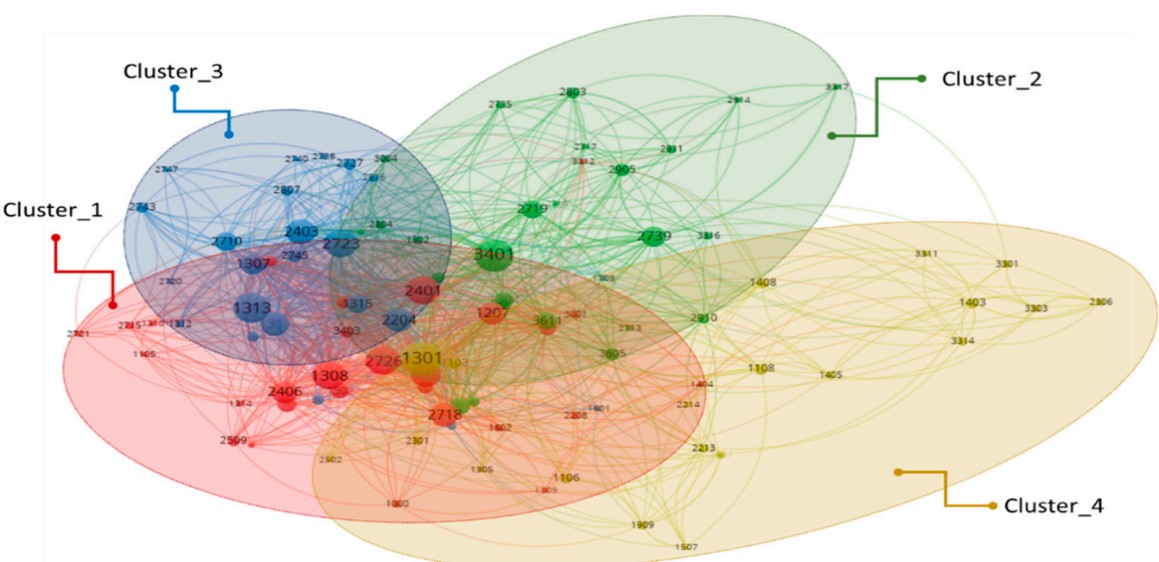

**Figure 10.** Clusters of interdisciplinary research on infectious diseases funded by the EU.

The Erasmus Universitair Medisch Centrum Rotterdam's research, "Early Detection of Emerging Viruses by Next Generation in Situ Hybridization", collaborated with multiple disciplines such as Food Animals (3403), Biochemistry, medical (2704), Immunology and Microbiology (miscellaneous) (2401), Medical Laboratory Technology (3607) and Virology (2406), and spent USD 205,891 between 2014–2016. In case of "New Millidrop Analyzer Miniaturized and Faster Clinical Microbiology Testing in Only One Drop" of the Millidrop Instruments SAS, it studied interdisciplinary approaches including Microbiology (medical) (2726), Applied Microbiology and Biotechnology (2402), Clinical Biochemistry (1308), Bioengineering (1502) and Medical Laboratory Technology (3607), with expenditures totalling USD 83,572 from 2017–2018. "Scalable Inference Algorithms for Bayesian Evolutionary Epidemiology" of the University of Oslo were relevant to the disciplines of Immunology and Microbiology (miscallaneous) (2401), Health Informatics (2718), Management Information Systems (1404), History and Philosophy of Science (1207) and Electrical and Electronic Engineering (2208), with expected expenditures of USD 2,924,954 during the time period of 2017–2022. "Swine Diseases Field Diagnostics Toolbox" of the Cy.R.I.C Cyprus Research and Innovation Center Ltd. encompasses Clinical Biochemistry (1308), Biochemistry, medical (2704), Microbiology (medical) (2726), Food Animals (3403), and Medical

Laboratory Technology (3607) with expected expenditures of USD 4,042,309 between 2017–2021, and "Virulence Factors of Facultative Pathogens and their Role outside the Human Host" of the Ecole Polytechnique Federale De Lausanne work in collaboration with the disciplines of Immunology and Microbiology (miscellaneous) (2401), Physiology (1314), Virology (2406), Ecology, Evolution, Behavior and Systematics (1105) and Hepatology (2721), with a budget of USD 1,987,741 during the time period of 2013–2018.

### 3.2.2. Public Health Policy and Infrastructure for the Prevention and Treatment of Infectious Diseases (Cluster 2)

Fourteen projects concerning public health for the prevention and treatment of infectious diseases have been conducted with a total budget of USD 62,382,324, which principally studied Health Policy (2719) and Community and Home Care (2905), with other 12 heterogeneous research fields such as Occupational Health (2739), Issues, ethics and legal aspects (2910), Cultural Studies (3316) and Pharmacy (3611) (see Table 7).

**Table 7.** Public health for the prevention and treatment of infectious diseases (Cluster 2).

| No. | COST (USD) | ORG_NAME | ASJC_CODE_5 | START_DATE | END_DATE | TITLE |
|---|---|---|---|---|---|---|
| 2 | 2,865,869 | King's College London | 2717;3317;2739;2905;2914 | 2014-08-01 | 2019-07-31 | 10/66 Ten Years on Monitoring and Improving Health Expectancy by Targeting Frailty Among Older People in Middle Income Countries |
| 2 | 12,826,010 | Medicines and Healthcare Products Regulatory Agency | 3401;2736;3611;2719;2905 | 2013-11-01 | 2017-10-31 | European Research Infrastructures for Poverty Related Diseases |
| 2 | 12,401,992 | European Vaccine Initiative—EEIG | 3401;2719;2905;2910;3316 | 2017-05-01 | 2022-04-30 | European Vaccine Research and Development Infrastructure |
| 2 | 34,130,994 | The University of Edinburgh | 2739;3401;2719;2910;2713 | 2017-01-01 | 2021-12-31 | Respiratory Syncytial Virus Consortium in Europe |

With regard to age-related diseases, the King's College London has conducted research under the title of "10/66 Ten Years on Monitoring and Improving Health Expectancy by Targeting Frailty Among Older People in Middle Income Countries" spending USD 2,865,869 between the 2014–2019 from multiple perspectives such as Geriatrics and Gerontology (2717), Demography (3317), Public Health, Environmental and Occupational Health (2739), Community and Home Care (2905) and Medical-Surgical (2914). In the case of poverty-related diseases, the Medicines and Healthcare products Regulatory Agency completed the project of "European Research Infrastructures for Poverty Related Diseases" with a budget of USD 12,826,010 during the time period of 2013–2017, which included heterogeneous disciplines such as Veterinary (miscellaneous) (3401), Pharmacology (medical) (2736), Pharmacy (3611), Health Policy (2719) and Community and Home Care (2905). For respiratory system-related diseases, the project, "Respiratory Syncytial Virus Consortium in Europe", researched by the University of Edinburgh is slated to spend USD 34,130,994 during the time period of 2017–2021, which covers multiple disciplines such as Public Health, Environmental and Occupational Health (2739), Veterinary (miscellaneous) (3401), Health Policy (2719), Issues, ethics and legal aspects (2910) and Epidemiology (2713).

### 3.2.3. Immunological Studies on Infectious Diseases (Cluster 3)

Immunological studies on infectious diseases comprising 6 projects, worth a total of USD 11,357,222, conducted chiefly based on Cell Biology (1307), Molecular Medicine (1313) and Immunology and

Allergy (2723), while adopting 12 different disciplines such as Developmental Biology (1309), Structural Biology (1315), Immunology (2403), Embryology (2710) and so on (see Table 8).

**Table 8.** Immunological studies on infectious diseases (Cluster 3).

| No. | COST (USD) | ORG_NAME | ASJC_CODE_5 | START_DATE | END_DATE | TITLE |
|---|---|---|---|---|---|---|
| 3 | 1,751,490 | Tel Aviv University | 1309;1313;1315;2710;2727 | 2015-07-01 | 2020-06-30 | A Genetic View of Influenza Infection |
| 3 | 1,754,713 | Universitaet Zuerich | 1313;1307;2807;1103;1315 | 2016-06-01 | 2021-05-31 | How Infection History Shapes the Immune System Pathogen-induced Changes in Regulatory T-Cells |
| 3 | 114,341 | Cardiff University | 1307;2723;1312;2403;2743 | 2017-09-01 | 2018-08-31 | Innate-Like T-Cells in Sepsis (ILTIS) Implications for Early Diagnosis and Rescue of Immune Suppression. |
| 3 | 1,754,861 | Academisch Ziekenhuis Groningen | 1307;2710;2916;2723;2743 | 2017-04-01 | 2022-03-31 | The Role of the Virome in Shaping the Gut Ecosystem during the First Year of Life |
| 3 | 3,653,371 | Universita' Degli Studi Di Milano-Bicocca | 2723;1307;2403;1313;2743 | 2015-01-01 | 2018-12-31 | Toll-Like Receptor 4 Activation and Function in Diseases: an Integrated Chemical-biology Approach |
| 3 | 2,328,446 | Medical Research Council | 1313;2204;2727;1309;1311 | 2017-10-01 | 2022-09-30 | Understanding the Complexity and Architecture in Protein Ubiquitination |

The Universitaet Zuerich, the Tel Aviv University, and the Medical Research Council played important roles in accomplishing the projects "How Infection History Shapes the Immune System Pathogen-induced Changes in Regulatory T-Cells" (USD 1,754,713 between 2016–2021) with Molecular Medicine (1313), Cell Biology (1307), Endocrine and Autonomic Systems (2807), Animal Science and Zoology (1103), Structural Biology (1315)), "A Genetic View of Influenza Infection " (USD 1,751,490 between 2015–2020) with Developmental Biology (1309), Molecular Medicine (1313), Structural Biology (1315), Embryology (2710), and Nephrology (2727), and "Understanding the Complexity and Architecture in Protein Ubiquitination" (USD 2,328,446 between 2017–2022) with Molecular Medicine (1313), Biomedical Engineering (2204), Nephrology (2727), Developmental Biology (1309) and Genetics (1311), respectively.

### 3.2.4. Development of Vaccine and Vaccine-Related Products (Cluster 4)

The development of vaccines and vaccine-related products comprised 4 projects worth USD 14,780,426, and was conducted chiefly based on Biochemistry, Genetics and Molecular Biology (miscellaneous) (1301) while adopting 11 different disciplines such as Business and International Management (1403), Management of Technology and Innovation (1405), Microbiology (2404), Parasitology (2405) and so on (see Table 9).

For instance, the project, "Leveraging Pharmaceutical Sciences and Structural Biology Training to Develop 21st Century Vaccines", from the University of Strathclyde was carried out with a budget of USD 1,243,357 during the time period of 2016–2020 comprising various disciplines such as Biomaterials (2502), Biochemistry, Genetics and Molecular Biology (miscellaneous) (1301), Biochemistry, Genetics and Molecular Biology (miscellaneous) (1301), Microbiology (2404), Infectious Diseases (2725) and Oncology (2730).

**Table 9.** Development of vaccine and vaccine-related products (Cluster 4).

| No. | COST (USD) | ORG_NAME | ASJC_CODE_5 | START_DATE | END_DATE | TITLE |
|---|---|---|---|---|---|---|
| 4 | 1,243,357 | University of Strathclyde | 2502;1301;2404;2725;2730 | 2016-06-01 | 2020-05-31 | Leveraging Pharmaceutical Sciences and Structural Biology Training to Develop 21st Century Vaccines |
| 4 | 3,057,009 | Universita degli Studi di Trento | 1301;1405;1403;2730;3002 | 2014-06-01 | 2019-05-31 | Outer Membrane Vesicles OMVs from Vaccinobacter: A Synthetic Biology Approach for Effective Vaccines against Infectious Diseases and Cancer |
| 4 | 1,506,843 | University College Cork, National University of Ireland, Cork | 1301;2741;2404;2502;2725 | 2014-10-01 | 2018-09-30 | Vaccines and Imaging Partnership |
| 4 | 8,973,217 | Academisch Medisch Centrum—Universiteit Van Amsterdam | 1301;2404;2405;2724;2725 | 2013-10-01 | 2017-09-30 | Developing and Testing a Novel Low-cost Effective Hookworm Vaccine to Control Human Hookworm Infection in Endemic Countries |

### 3.3. Comparison between the US and EU

The US and EU shared the same interest in public health policy related to infectious diseases, which stems from the results of Public health for HIV-vulnerable group/Respiratory health for children/HR for infectious diseases (Cluster 1) of the US and Public health for the prevention and treatment of infectious diseases (Cluster 2) of the EU. However, the target group of the interdisciplinary researches of the two were different. The US focused on vulnerable members of society, while the EU studied the benefits of vaccination among regular citizen. Furthermore, while the subject of the epidemic prevention and policy research in the US is clear and detailed, from a whole social and economic structure of viewpoint developing infrastructure accounted for the majority of the European Unions' projects.

From the results of diagnosis and treatment of infectious diseases by using advanced technology (Cluster 2) of US and Public health for the prevention and treatment of infectious diseases (Cluster 2) of EU, they have both heavily invested in research and development of epidemic prevention and treatment. Especially research on the diagnosis of infectious diseases has been actively conducted by using advanced technologies such as ICT, data, and platform. Unlike the EU, which focused their immunological studies on infectious diseases, the US immunological studies mainly concentrated on inflammatory diseases.

When taking a closer look at biological studies of the mechanism of inflammatory diseases caused by infectious diseases and the development of therapies for them (Cluster 3) of US and Immunological studies on infectious diseases (Cluster 3) of the EU, common biological and immunological approaches to the pathogenesis and treatment of epidemics is likely to break down in terms of pathogens and out broken organs of infectious diseases. Likewise, depending on examination of clinical trials of vaccines and products to help treat and prevent infectious diseases (Cluster 5) of US and Development of vaccine and vaccine-related products (Cluster 4) of the EU, the US shares the same goal as the EU to develop, manage, and evaluate vaccine-related products.

From investigating the strengthening research capacity for epidemiology (Cluster 4) of the US, many projects have invested in developing human resources in epidemiology. Supporting research laboratories and performing the role of government health departments.

## 4. Discussion and Conclusions

The aim of this study was to understand the trends of research on infectious diseases that globally endangering human health and well-being from an interdisciplinary approach since mid the 2010s, thereby deducing strategic directions for governmental R&D. In particular, two leading

scientific funding organisations or programmes that played dominant roles in infectious diseases were investigated. According to the results, the US has invested in five interdisciplinary research areas (13 sub-clusters) of infectious diseases with a total budget of USD 179,170,346 and with 333 projects, which contained 118 heterogeneous disciplines. On the other hand, the EU has funded 19 projects which were worth about USD 87,876,000 and have experimented with 54 different disciplines. In summary, the US has significantly invested more in interdisciplinary research of infectious diseases than the EU since the mid-2010s. Although it is hard to directly compare the research fields of both nations in terms of discipline, four characteristics they had in common in terms of interdisciplinary research on infectious diseases was identified as follows: public health policy of infectious diseases, epidemic prevention and treatment, segmentations of biological and immunological approaches to the pathogenesis and treatment of epidemics, and vaccine development.

The EU heavily concentrated on public health for infectious diseases or prevention, diagnosis and treatment of infectious diseases (see Cluster 2 of the EU (USD 62.4 million) and Cluster 1 of the US (USD 9.3 million) or Cluster 2 of the US (USD 18.3 million)). It presumed that the Horizon 2020 programmes of the EU encouraged strong involvement of the EU member nations to support mechanisms for contributing to the establishment of a research ecosystem of wider reach and benefit in the EU [17].

Regarding the segmentation of biological and immunological studies, the US (Cluster 3, USD 37,588,973) invested more heavily than the EU (Cluster 3, USD 11,357,222) via a variety of projects. Particularly, the US has adopted more diverse knowledge stemming from multiple disciplines including Biochemistry, Genetics and Molecular Biology (miscellaneous) (1301), Cancer Research (1306), Cell Biology (1307), Cellular and Molecular Neuroscience (2804), Animal Science and Zoology (1103), and Applied Microbiology and Biotechnology (2402). On the contrary, the EU has focused on the vaccine development more than the US (see Cluster 4 of the EU (USD 4.9 million) and Cluster 5 of the US (USD 0.8 million)). As a result, research areas from the EU have included more distinctive disciplines such as Agronomy and Crop Science (1102), Horticulture (1108), Biochemistry, Genetics and Molecular Biology (miscellaneous) (1301), Global and Planetary Change (2306) and so on. The University of Strathclyde in the UK has taken the lead the interdisciplinary research on vaccine development of infectious diseases since 2016. It allows other infectious disease-related research organisations to recognise its specialised disciplines, which may offer a hint for coordinating the work of joint researches.

For the US, it has reinforced its capacity for epidemiology through significant investment (Cluster 4 of the US (USD 113,924,725)). Therefore, it is imperative that nations or organisations outside of the US review the research results from organisations in the US before considering launching a research programme.

**Author Contributions:** Conceptualization, K.K.; methodology, Y.H.; software, Y.H.; validation, K.K., J.K. and Y.H.; formal analysis, Y.H.; investigation, K.K. and J.K.; data curation, J.K. and Y.H.; writing—original draft preparation, K.K. and Y.H.; writing—review and editing, K.K., J.K. and Y.H.; visualization, Y.H.; supervision, K.K. and J.K.

**Funding:** This research received no external funding.

**Conflicts of Interest:** The authors declare no conflict of interest.

# Appendix A

**Table A1.** Interdisciplinary research fields of infectious diseases in the US.

| Cluster No. | Sub-Cluster No. | # of ASJC Codes in Sub-Cluster | Component ASJC Code (Main) | Component ASJC Code (Coordinated) | # of R&D Projects | The Total Cost of R&D Projects ($) | Major Organisation Performing Projects |
|---|---|---|---|---|---|---|---|
| 1. Public health for HIV-vulnerable group/Respiratory health for children/HR for infectious diseases | 1-1 HIV | 7 | Health Policy (2719) Nursing (miscellaneous) (2901) Oncology(nursing) (2917) Psychiatric Mental Health (2921) | Social Psychology (3207) Health (social science) (3306) Communication (3315) | 34 | 4,351,252 | University of North Carolina Chapel Hill Duke University University of Minnesota Twin Cities University of Missouri-Columbia University of Chicago Yale University Infectious Diseases Institute University of Wisconsin Madison |
| | 1-2 Heath assessment of children | 5 | Medicine (miscellaneous) (2701) Epidemiology (2713) Obstetrics and Gynaecology (2729) | Public Health, Environmental and Occupational Health (2739) Radiation (3108) | 2 | 3,530,344 | University of Utah |
| | 1-3 Research scholar and educational programs | 9 | Earth-Surface Processes (1904) Paediatrics (2919) Review and Exam Preparation (2923) Pharmacology, Toxicology and Pharmaceutics (miscellaneous) (3001) | Social Sciences (miscellaneous) (3301) Education (3304) Health Professions (miscellaneous) (3601) Complementary and Manual Therapy (3603) Emergency Medical Services (3604) | 10 | 1,389,374 | Colorado State University-Fort Collins Tufts University Boston |
| 2. Diagnosis and treatment of infectious diseases by using advanced technology | 2-1 Information technology-based infectious disease diagnosis | 15 | Management Information Systems (1404) Computer Science (miscellaneous) (1701) Computational Theory and Mathematics (1703) Computer Networks and Communications (1705) Hardware and Architecture (1708) Human-Computer Interaction (1709) Information Systems (1710) Signal Processing (1711) | Space and Planetary Science (1912) Biomedical Engineering (2204) Electrical and Electronic Engineering (2208) Computational Mathematics (2605) Health Informatics (2718) Health Information Management (3605) Radiological and Ultrasound Technology (3614) | 12 | 12,041,079 | The University of Pittsburgh Cornell University Ithaca Advanced Microsensors Corporation City College of New York Georgia Tech Research Corporation Research Triangle Institute University of Oklahoma Norman University of Utah |
| | 2-2 Molecular biology-based diagnosis and treatment | 13 | Biochemistry (1303) Biophysics (1304) Biotechnology (1305) Physiology (1314) Structural Biology (1315) Bioengineering (1502) Inorganic Chemistry (1604) | Organic Chemistry (1605) Spectroscopy (1607) Polymers and Plastics (2507) Statistics and Probability (2613) Drug Discovery (3002) Pharmaceutical Science (3003) | 19 | 5,532,582 | Washington University The Johns Hopkins University Nubad Llc University of Arizona University of Michigan At Ann Arbor University of Rochester Adjuvance Technologies Inc University of Oklahoma Norman Zata Pharmaceuticals Inc |
| | 2-3 Wastewater treatment for preventing infectious diseases | 5 | Fluid Flow and Transfer Processes (1507) Process Chemistry and Technology (1508) Geotechnical Engineering and Engineering Geology (1909) | Ecological Modelling (2302) Management, Monitoring, Policy and Law (2308) | 1 | 332,441 | Rutgers State University of New Jersey—New Brunswick |
| | 2-4 Eye disease stemmed from infectious diseases | 5 | Strategy and Management (1408) Management Science and Operations Research (1408) Engineering (miscellaneous) (2201) | Cardiology and Cardiovascular Medicine (2705) Oral Surgery (3504) | 1 | 399,420 | University of California San Francisco |

**Table A1.** *Cont.*

| Cluster No. | Sub-Cluster No. | # of ASJC Codes in Sub-Cluster | Component ASJC Code (Main) | Component ASJC Code (Coordinated) | # of R&D Projects | The Total Cost of R&D Projects ($) | Major Organisation Performing Projects |
|---|---|---|---|---|---|---|---|
| 3. Biological studies on the mechanism of inflammatory diseases caused by infectious diseases and the development of therapies | 3-1 Inflammation treatments caused by infectious diseases | 20 | Biochemistry, Genetics and Molecular Biology (miscellaneous) (1301) /Cancer Research (1306) Cell Biology (1307) Molecular Biology (1312) Molecular Medicine (1313) Immunology (2403) Embryology (2710) Gastroenterology (2715) Genetics(clinical) (2716) Hepatology (2721) | Immunology and Allergy (2723) Nephrology (2727) Otorhinolaryngology (2733) Physiology (medical) (2737) Reproductive Medicine (2743) Rheumatology (2745) Transplantation (2747) Endocrine and Autonomic Systems (2807) Toxicology (3005) Periodontics (3506) | 103 | 31,226,545 | University of Colorado Denver H Lee Moffitt Cancer Center & Research Institute Harvard School of Public Health University of North Carolina Chapel Hill University of Washington Dartmouth College Massachusetts General Hospital The University of Pittsburgh Harvard University Keystone Symposia. Etc. |
| | 3-2 sleepy sickness | 5 | Insect Science (1109) Developmental Biology (1309) Genetics (1311) | Ecology (2303) Cellular and Molecular Neuroscience (2804) | 4 | 416,250 | Yale University |
| | 3-3 viral diseases in human and animals | 10 | Animal Science and Zoology (1103) Environmental Science (miscellaneous) (2301) Immunology and Microbiology (miscellaneous) (2401) Applied Microbiology and Biotechnology (2402) Parasitology (2405) | Virology (2406) Infectious Diseases (2725) Microbiology (medical) (2726) Veterinary (miscellaneous) (3401) Food Animals (3403) | 24 | 5,946,178 | Duke University Stanford University Arisan Therapeutics Inc Auburn University At Auburn Emory University University of California DavisUniversity of Pennsylvania Agricultural Research Service Broad Institute Inc Texas A & M University University of Maryland College Pk Campus University of Rochester University of Washington |

**Table A1.** *Cont.*

| Cluster No. | Sub-Cluster No. | # of ASJC Codes in Sub-Cluster | Component ASJC Code (Main) | Component ASJC Code (Coordinated) | # of R&D Projects | The Total Cost of R&D Projects ($) | Major Organisation Performing Projects |
|---|---|---|---|---|---|---|---|
| 4. Strengthening research capacity for epidemiology | 4-1 small animal models for infectious diseases | 9 | Horticulture (1108) Arts and Humanities (miscellaneous) (1201) History and Philosophy of Science (1207) Computer Graphics and Computer-Aided Design (1704) | Computer Science Applications (1706) Software (1712) Mathematics (miscellaneous) (2601) Complementary and alternative medicine (2707) Pathology and Forensic Medicine (2734) | 19 | 8,688,987 | Mount Sinai School of Medicine Utah State University Georgetown University Public Health England Children's Hospital Medical Center Cincinnati Kmt Hepatech Inc Pennsylvania State Univ Hershey Med Ctr Southern Research Institute University of Georgia University of Texas Medical Br Galveston Upstate Medical University |
| | 4-2 epidemiology and health system | 10 | Agricultural and Biological Sciences (miscellaneous) (1011) History (1202) Clinical Biochemistry (1308) Biochemistry, medical (2704) Endocrinology, Diabetes and Metabolism (2712) | Family Practice (2714) Internal Medicine (2724) Fundamentals and skills (2908) Medical-Surgical (2914) Medical Laboratory Technology (3607) | 102 | 105,235,738 | Arizona State Department of Health Services New York City Health/Mental Hygiene Virginia State Department of Health Georgia State Department of Public Health Indiana State Department of Health Kansas State Department of Health and Environment Los Angeles Cnty Off of Aids Progs & Pol Louisiana State Office of Public Health Maine State Dept/Health/Human Servs Minnesota State Dept of Health Etc |
| 5. Clinical trials on vaccines and products to help treat and prevent infectious diseases | 5-1 Clinical trials on vaccines and products to help treat and prevent infectious diseases | 5 | Industrial relations (1410) Critical Care and Intensive Care Medicine (2706) Oncology (2730) | Pharmacology (nursing) (2920) Pharmacy (3611) | 2 | 80,156 | Saint Louis University University of Iowa |

**Table A2.** Interdisciplinary research fields of infectious diseases in the EU.

| Cluster No. | # of R&D Projects | The Total Cost of R&D Projects ($) | Major Organisation Performing Projects | # of ASJC Codes | Component ASJC Code | |
|---|---|---|---|---|---|---|
| 1. Detection and profiling for pathogens | 5 | 9,244,467 | Cy.R.I.C Cyprus Research and Innovation Center Ltd. Ecole Polytechnique Federale de Lausanne Erasmus Universitair Medisch Centrum Rotterdam Millidrop Instruments SAS Universitetet i Oslo | 16 | Ecology, Evolution, Behavior and Systematics (1105) Clinical Biochemistry (1308) Physiology (1314) Management Information Systems (1404) Bioengineering (1502) Electrical and Electronic Engineering (2208) Immunology and Microbiology (miscellaneous) (2401) Applied Microbiology and Biotechnology (2402) | Parasitology (2405) Virology (2406) Biochemistry, medical (2704) Health Informatics (2718) Hepatology (2721) Microbiology (medical) (2726) Food Animals (3403) Medical Laboratory Te chnology (3607) |
| 2. Public health for the prevention and treatment of infectious diseases | 5 | 62,382,324 | Medicines and Healthcare Products Regulatory Agency European Vaccine Initiative – EEIG King's College London University of Edinburgh University College London | 14 | Epidemiology (2713) Family Practice (2714) Geriatrics and Gerontology (2717) Health Policy (2719) Pharmacology (medical) (2736) Public Health, Environmental and Occupational Health (2739) Biological Psychiatry (2803) | Community and Home Care (2905) Issues, ethics and legal aspects (2910) Medical-Surgical (2914) Cultural Studies (3316) Demography (3317) Veterinary (miscellaneous) (3401) Pharmacy (3611) |
| 3. Immunological studies on infectious diseases | 6 | 11,357,222 | Academisch Ziekenhuis Groningen Cardiff University Medical Research Council Tel Aviv University Universita' Degli Studi di Milano-Bicocca Universitaet Zuerich | 15 | Animal Science and Zoology (1103) Cell Biology (1307) Developmental Biology (1309) Genetics (1311) Molecular Biology (1312) Molecular Medicine (1313) Structural Biology (1315) Biomedical Engineering (2204) | Immunology (2403) Embryology (2710) Immunology and Allergy (2723) Nephrology (2727) Reproductive Medicine (2743) Endocrine and Autonomic Systems (2807) Nutrition and Dietetics (2916) |
| 4. Development of vaccine and vaccine-related products | 3 | 4,892,410 | Universita Degli Studi di Trento Universitair Medisch Centrum Utrecht Vitamfero SA | 9 | Agronomy and Crop Science (1102) Horticulture (1108) Biochemistry, Genetics and Molecular Biology (miscellaneous) (1301) Business and International Management (1403) Management of Technology and Innovation (1405) | Environmental Science (miscellaneous) (2301) Global and Planetary Change (2306) Biomaterials (2502) Development (3303) |

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
