# Peer review of "National Scientific Funding for Interdisciplinary Research: A Comparison Study of Infectious Diseases in the US and EU"

_sustainability, doi:10.3390/su11154120_

Round 1

Reviewer 1 Report

I think the method is quite nice and the study is novel. However, due to large number of grammatical errors, the manuscript needs to be edited by someone proficient in English writing for correct English usage and then rereviewed by authors. The way paper is written it's not acceptable English. The content is good though.

Author Response

Response to Reviewer 1 Comments

Point 1: I think the method is quite nice and the study is novel. However, due to large number of grammatical errors, the manuscript needs to be edited by someone proficient in English writing for correct English usage and then reviewed by authors. The way paper is written it's not acceptable English. The content is good though.

Response 1: We conducted the English proofreading and revision of the manuscript by asking experts. We will attach the revised manuscript.

Reviewer 2 Report

The manuscript entitled: "National scientific funding for interdisciplinary research: a comparison study of infectious diseases in the US and EU" reportd information on research trends with reference to United States and Europe. One remark regards the Abstract: please correct leading countries, EU is European Union not a single Country, while United States can be considered a Country (made of different States).

The methodology used, namely Scientometrics examines the quantitative features and characteristics of science and scientific research studied by statistical mathematical methods. The approach is appropriate and well described. The subject is interesting and the paper is well written . Some minor revision before the acceptance:

 able  1 should be changed into Figure.

In  Introduction the authors should mention and mark the importance of studies of use of pharmaceuticals as well as communication strategies for proper information for achieving an healthy ageing. In this regards proper reference should be mentioned. A few lines referred to communication,  compliance and differences in conditions between men and women should be added. Some References shoule be added to better exploit the context. In the following some are listed and should be added to the text:

Putignano D, Bruzzese D, Orlando V, Fiorentino D, Tettamanti A, Menditto E. Differences in drug use between men and women: an Italian cross sectional study. BMC Womens Health. 2017; 17(1):73.

Scala D, Menditto E, Armellino MF, Manguso F, Monetti VM, Orlando V, Antonino A, Makoul G, De Palma M. Italian translation and cultural adaptation of the communication assessment tool in an outpatient surgical clinic. BMC Health Serv Res. 2016; 16:163.

Menditto E, Cahir C, Aza-Pascual-Salcedo M, Bruzzese D, Poblador-Plou B, Malo S, Costa E, González-Rubio F, Gimeno-Miguel A, Orlando V, Kardas P, Prados-Torres A. Adherence to chronic medication in older populations: application of a common protocol among three European cohorts. Patient Prefer Adherence. 2018; 12: 1975-1987.

Menditto E, Orlando V, Coretti S, Putignano D, Fiorentino D, Ruggeri M. Doctors commitment and long-term effectiveness for cost containment policies: lesson learned from biosimilar drugs. Clinicoecon Outcomes Res. 2015; 7:575-581.

he section Material and Methods should be formatted following the guidelines of MDPI. Part of results (Tables and Figures) should be moved in a new section Supplementary Material. English phrasing/grammar/spelling should be checked and improved for better readability.

Author Response

Response to Reviewer 2 Comments
